# Why Talking Is Not Cheap: Adverse Events and Informal Communication

**DOI:** 10.3390/healthcare12060635

**Published:** 2024-03-12

**Authors:** Anthony Montgomery, Olga Lainidi, Katerina Georganta

**Affiliations:** 1Department of Psychology, Northumbria University, Newcastle upon Tyne NE1 8ST, UK; 2School of Psychology, University of Leeds, Leeds LS2 9JT, UK; o.lainidi@leeds.ac.uk; 3Department of Psychology, University of Ioannina, 451 10 Ioannina, Greece; k.georganta@uoi.gr

**Keywords:** healthcare, informal communication, gossip, employee silence, patient safety

## Abstract

Healthcare management faces significant challenges related to upward communication. Sharing information in healthcare is crucial to the improvement of person-centered, safe, and effective patient care. An adverse event (AE) is an unintended or unexpected incident that causes harm to a patient and may lead to temporary or permanent disability. Learning from adverse events in healthcare is crucial to the improvement of patient safety and quality of care. Informal communication channels represent an untapped resource with regard to gathering data about the development of AEs. In this viewpoint paper, we start by identifying how informal communication played a key factor in some high-profile adverse events. Then, we present three Critical Challenge points that examine the role of informal communication in adverse events by (1) understanding how the prevailing trends in healthcare will make informal communication more important, (2) explaining how informal communication is part of the group-level sensemaking process, and (3) highlighting the potential role of informal communication in “breaking the silence” around critical and adverse events. Gossip, as one of the most important sources of informal communication, was examined in depth. Delineating the role of informal communication and adverse events within the healthcare context is pivotal to understanding and improving team and upward communication in healthcare organizations. For clinical leaders, the challenge is to cultivate a climate of communication safety, whereby informal communication channels can be used to collect soft intelligence that are paths to improving the quality of care and patient safety.

## 1. Introduction

Learning from adverse events (AEs) in healthcare is crucial to the improvement of person-centered, safe, and effective patient care. But, is there something that AEs are telling us that we are reluctant to learn? In the following paper, we begin by outlining the importance of informal communication in the following AEs: Mid-Staffordshire NHS Foundation Trust Public Inquiry, the case of Dr. Jayant Patel Bundaberg Hospital, Australia, the scandal at the Bristol Royal Infirmary, and the Lucy Letby scandal. The most common response to adverse events has been to increase the layers of formality to the healthcare process as a way to reduce the probability of critical events being missed. However, in this paper, we will argue that informal communication (which increases due to extra formality) should be our area of focus if we want to better understand how information is shared and communicated long before a critical point is reached. We will analyze informal communication via two very common and interrelated paths in healthcare—employee silence and gossip. We will argue that employee silence does not always equate with silence in the organization, and can find expression in the form of gossip that represents an ‘early warning system’ for adverse events [1]. The informal communication channels that individuals use are an untapped source of information about adverse events. Analyzing informal communication channels helps us understand the tension that exists between care and confidentially, which are the two key values in healthcare. We have structured the paper to outline what we can learn from adverse events, followed by an outline of three Critical Challenge points that can inform future research.

## 2. A Note on Definitions

Prior to beginning our discussion, a short note on definitions is warranted. Our discussion of adverse events will focus on the role of informal communication in healthcare settings, meaning “brief, more opportunistic interactions, taking place within and between different healthcare team members, as well as with a patient’s family members or other caregivers” [2] (p. 289). Informal communication in work settings is defined as voluntary talk that does not have to be solely work- or task-focused [3]. There are different types of informal communication, but the present paper will explore the phenomenon of gossip in detail. Gossip is a form of informal communication “at the core of human social relationships” [4] (p. 811) and is defined as “the process of informally communicating value-laden information about members of a social setting” [5] (p. 25). Gossip occurs amongst at least two people and includes evaluative talk about those not present [6,7,8,9]. It is an informal way of communicating rules and establishing norms that lead to information and risk sharing [5]. Evaluations can be of either positive or negative valence and are more likely to be shared with members of the ingroup rather than the outgroup [10]. Senders and receivers report gossip with positive, negative, and neutral valence, each representing roughly a third of the gossip instances [11]; however, definitions generally do not attach valence or morality to its content or consequences [12,13,14,15]. Because the content of gossip can be both negative and positive, it has the potential to build both positive and negative social reputations [16,17,18,19].

Finally, sensemaking will be discussed throughout the paper. “Sensemaking involves the ongoing retrospective development of plausible images that rationalize what people are doing” [20] (p. 409). Moreover, efforts at sensemaking are more intense when the current state of the world is perceived to be different from the expected state of the world, which dovetails with the occurrence of AEs in healthcare. The emphasis in sensemaking on both understanding flux/chaos and the need to communicate makes it the perfect framework with which to disentangle employee silence and informal communication (i.e., gossip).

## 3. Adverse Events (AEs) and Informal Communication

An adverse event (AE) is an unintended or unexpected incident that causes harm to a patient and may lead to temporary or permanent disability [21]. Unless the protocol states otherwise, these must always be recorded on a case report form (CRF) and in the patient’s medical notes. It is internationally recognized that 10–25% of healthcare episodes (in general hospitals, community hospitals, and general practice) are associated with an adverse event [22]. As learning from adverse events (AEs) is crucial to the improvement of person-centered, safe, and effective patient care [23], we draw upon examples of AEs to demonstrate how informal communication becomes an integral part of the sensemaking process [20] that, in some cases, can affect the decision to withhold concerns or speak up. We view informal communication as a collective attempt to make sense of problematic situations and what this can tell us about information sharing and transparency in the healthcare sector. Such communication is more likely to go under the radar and is more difficult to access in terms of research. Stress increases the likelihood of formal silence. There is significant evidence that healthcare education is building a culture of performance first, where individual well-being and asking difficult questions are far down the list of priorities [24]. A ‘performance-first’ culture does not encourage speaking up. Conversely, a ‘performance-first’ culture can encourage “speaking about” through gossip and informal conversations among colleagues, particularly when it comes to superiors and when speaking up against leaders is not an option.

Inquiries into the failings of care over time have found that, during the periods under investigation, the informal raising of concerns by many staff and managers about the level of care provided to patients was significantly increased [25]. For example, the Mid-Staffordshire Foundation NHS Trust failings of care highlighted how poor care was associated with the avoidable deaths of patients [26] (for details, see the Executive Summary of the Report of the Mid-Staffordshire NHS Foundation Trust Public Inquiry). The first report [27] clearly stated that there needed to be more openness and transparency throughout the trust, as, for example, incidents of poor patient care delivery were not formally recorded in the system or communicated to commissioners and regulators.

We know that the role of whistleblowing in exposing the failings and cover-ups was critical [28]. Before that, repetitive care declines were part of the employees’ informal communications. The widespread disengagement in leadership responsibility addressed in the second report by Francis [29] describes failings and AEs of varying severity that can in no way go under the gossip radar [4]. The fact that the Francis report consists of two volumes suggests the magnitude and the multitude of AEs during the period under investigation. Unfortunately, informal communication routes in the forms of gossip and rumor around the series of failures, inadequacies, and cover-ups were not powerful enough to create a force to counteract what, in the Francis report, was named “negative culture” [29]. In their signals analysis paper, Carman et al. [30] highlighted how the content of informal communication amongst staff until May 2009 was indicative of what was happening, with employees discussing how they would not want their relatives and themselves to be patients. Thus, what employees informally talk about can be an important aspect of soft intelligence, a valuable resource with which to monitor how the day-to-day patient care is going, and probably a metric of care delivery quality—which means there are positive functions to gossip, as the pool of data related to what happens in the reality of the day-to-day delivery of patient care. The potential of gossip to be a necessary form of soft intelligence is highlighted by the recent typology of gossip by Lee and Barnes [31] into four categories, which include (a) protection-based gossip that alerts the workgroup to potential threats, (b) derogation-based gossip that negatively influences a coworker’s reputation, (c) endorsement-based gossip that enhances a coworker’s reputation through praise of exemplary behavior, and (d) communion-based gossip that strengthens interpersonal ties through social enjoyment. This approach is particularly relevant for adverse health and social care events in that it focuses on the behavior of consequence for the gossip sender’s relationships with others.

The inquiry in the case of Dr. Jayant Patel at Bundaberg Hospital, Australia [32] can also provide insight into how employees responded to failings in patient care over time. Once again, the role of whistleblowing here was crucial. Still, an analysis of the events that led to the inquiry shows that collective sensemaking occurred before any whistleblowing decisions. On the level of informal communication, employees were unofficially discussing their concerns during the period of sensemaking, which led to either silence or formal reporting or whistleblowing [32]. When adverse events started occurring, employees engaged in discussions regarding Dr. Patel’s behaviors, performance, and decisions amongst themselves since the first months after his arrival. We could hypothesize that these discussions—at that time, with no evidence and no severe AEs happening—can be classified as gossip, where a group of employees discussed Dr. Patel “behind his back”. However, following the events that led to the formal inquiry (for details, see Edwards et al.) [32], it becomes more and more clear how the initial informal communication in the form of gossiping about the newly arrived surgeon is the unofficial attempt of the staff to make sense of decisions made by the hospital leaders. Testimonies provided by employees have indicated that Dr. Patel’s behavior became a frequent topic of discussion. Edwards et al. [32] identified that those informal discussions were part of an attempt at “sensemaking” at this stage. While we do not imply any causal relationships between gossip and disclosure, it is possible that “keeping the gossip about Dr. Patel alive” for 24 months contributed to the external whistleblowing—after several attempts of official and unofficial reporting internally. The strategic benefit of temporarily remaining silent has been highlighted by Stouten et al. [33] in that it can serve several strategic functions, such as waiting for a more strategic time to act and allowing for time to form alliances to formulate an effective response.

Most recently, the Lucy Letby case has highlighted the problems inherent in a no-blame culture [34]. In 2023, Lucy Letby, a neonatal nurse, was sentenced to life in prison for the murder of seven babies and the attempted murder of six others. Much of the commentary on the case focused on the fact that concerns were raised by healthcare staff but ignored by managers [35]. The UK government has ordered an independent public inquiry into the circumstances surrounding the murders, so it is too early to reach any meaningful conclusions. However, the various reports surrounding the case reveals the importance of informal communication, such as “It was no secret that Letby was present when the infants suddenly collapsed, yet her crimes were so subtle they were imperceptible. Trainees started referring to her as “the angel of death”, …”. [36]. As noted by Leary [37], “success” in healthcare is measured by how much work is done, not how well the work is done. The Letby case, although arguably an outlier, is a reminder that the increased formalization of medical care is insufficient for gathering and hearing the important intelligence circulating in a hospital prior to an AE.

## 4. Understanding AEs via the Lens of Employee Silence/Voice

Healthcare workers frequently remain silent about work-related matters and prefer to use informal communication channels to share information. However, the evidence establishing the ubiquity of employee silence and voice in healthcare accounts largely for formal communication, upward communication channels, and speaking up to superiors and colleagues. The ongoing movement within healthcare towards standardizing procedures and digitalizing communications affirms that such informal routes of communication will become even more important. Accordingly, overburdened healthcare workers will continue seeking alternative ways to feel in control of their narratives and safely share sensitive information about work practices, and especially AEs.

Within the existing management practices, employee silence and voice are organizational behaviors that most frequently do not extend to informal communications between members of staff. In terms of our need to better understand AEs, this is lost data and a missed opportunity. For example, employees might be informally sharing their concerns in the form of venting or complaining about a colleague’s or superior’s behavior. When this is part of the “watercooler talk”, it is likely to go under the radar as gossip rather than reach formal channels of communication and lead to change. In the event of an AE, staff can keep quiet while also informally discussing important issues (e.g., safety issues or unprofessional behaviors) amongst themselves or with family/friends through informal channels. Choosing gossip over a formal voice route can potentially function as a collective process of sensemaking [38,39] and discharge through the channels of informal communication among employees, when formal reporting is not considered an option. Viewing gossip as an alternative way to not staying silent can contribute to a better understanding of how and why employees might choose to withhold their voice or refrain from speaking up—especially when feeling unprotected from the consequences of doing so.

Employee silence/voice and gossip at work have been studied in parallel research domains. Still, considerable overlap exists, especially concerning events that challenge staff well-being and patient safety. Interest in workplace gossip endures because communication among colleagues is a vital source of informal communication within healthcare teams [40]. Additionally, studying all forms of communication in healthcare is critical given the relationship between effective communication between team members and improved patient safety and better healthcare delivery outcomes [41], increased self-reported employee well-being [42], lower self-reported burnout levels [43], and, overall, a more positive organizational culture [44].

Waddington [4] maintained that the negative view of gossip has overshadowed its potential value for healthcare organizations, with positive gossip often being linked to desired outcomes such as better social bonds within the team [45]. In this paper, we focus on the potential positive functions of gossip in relation to employee silence in healthcare, amidst crises and adverse events. Gossip, whether positive or negative, is anathema to systems pushing towards formalization and professionalization in healthcare. Obviously, getting everything (i.e., decision and opinion) on the record is desirable, but there is the distinct possibility that such approaches can make staff defensive and selective about what goes on the record [46]. The emphasis on informal communication stems from literature identifying the persistent problems found in formal communication channels [47], the limited availability of research reporting the effectiveness of speaking-up interventions in the sector [48], and the ubiquitous nature of employee silence in healthcare [49]. In the following, we present three Critical Challenge points with which to advance our understanding of gossip and employee silence in healthcare, by (1) understanding how the prevailing trends in healthcare will make informal communication more important, (2) explaining how informal communication is part of the group-level sensemaking process, and (3) highlighting the potential role of informal communication in “breaking the silence” around critical and adverse events.

## 5. Challenge Points

### 5.1. Challenge Point 1: Informal Communication Channels Will Become Increasingly Important in Healthcare

Healthcare professionals report experiencing high levels of work-related stress and are among the professions with the highest reported levels of burnout [50,51]. Burnout is linked to a culture of “performance-first”, where mistakes are unwanted—even those that have minor implications for patient safety and quality of care, thus valorizing non-disclosure [49]. Moreover, healthcare organizations are constantly under formal scrutiny. Often, there is a tendency to attribute individual responsibility to the individual nurse and/or physician for suboptimal care delivery rather than identify organizational and systemic problems on a senior level [52,53]. This reality generates a culture of fear and blame, where everyone quickly becomes aware of the consequences of formal information sharing and speaking up. However, people need to share their thoughts and feelings, so informal sharing channels become an important outlet especially during intense events (e.g., AEs).

The workload and the demands of collaborating with many colleagues within and across teams expose healthcare professionals to a considerable amount of information related to day-to-day care, positive and negative events, concerns, and suspicions. This information overflow cannot only be managed via formal information-sharing channels due to this lack of trust. As informal communications are not likely to put healthcare professionals’ jobs at stake and limit liability, informal communication (e.g., gossip) becomes the most viable option as a dynamic conversational event [54].

To achieve high-quality care, healthcare staff must communicate effectively both formally and informally [55]. Healthcare professionals collaborate with colleagues in both the same (e.g., nurses collaborating with nurses) and different specialties (e.g., nurses collaborating with doctors; doctors collaborating with allied professionals). Most often, clinical teams consist of members with different positions in the hierarchy. This can sometimes lead to communication complications that formal routes of transmission cannot always help resolve [2]. Taking into account excessive workloads, and constraints in time and resources, along with the need for practical solutions, healthcare staff often relies on the so-called “workarounds” [56], whereby the staff develops creative solutions to resource and staff shortages. Thus, there is an acceptance that being “silent” about the gaps in care is practical and solution-focused [57,58]. Workarounds present an interesting paradox. On the one hand, there may be an underlying driver of professional shaming that could help explain the silence around them. On the other hand, it is an organic response to silence (as opposed to apathy): because change will not ever be forthcoming, the problem needs to be “worked around” [59]. Thus, informal communication is often critical [2].

Congruently, attempts to include more formal channels of communication (i.e., computerized provider order entry, patient portal systems, and instant messaging apps) to overcome the communication barriers in healthcare have proven to be much less helpful than expected. These approaches tend to “formalize” communication at work, making it even more procedural, reserved, and planned, and potentially creating a work environment where employees feel every word is monitored [60]. Doing so seriously harms the spontaneity and the dynamic character of day-to-day work communications. Moreover, they lessen employees’ feeling of control over their jobs, as aspects of their job become procedural, contributing to depersonalization and interpersonal distance between the team members [61].

Research has shown that informal communication among healthcare employees is positively linked to better support of the day-to-day needs and work demands of healthcare organizations [62,63]. Burm et al. [2] identified three valuable aspects of informal communication events in healthcare organizations: (1) a better sense of patients’ baseline function; (2) a better understanding of the patient’s needs, and (3) a better insight into the goals of care. Ward corridors and staff lounges are “knowledge-sharing venues”, contributing to safer clinical practice and patient safety [64]. However, it is still unclear if all types of “informal chats” bring the same value in improving teamwork and quality of care. For example, studies recognize that informal interactions between nurses and non-physician professionals were positively linked with better collaborative care [65]. Conversely, informal interactions between nurses and physicians did not promote positive outcomes [66]. Accordingly, although informal communication is a valuable resource, there is not yet enough evidence indicating when and how it can better serve patient safety goals and quality of care.

In terms of informal communication routes, the extensively studied phenomenon of gossip is worthy of special consideration. According to Fan et al. [67], gossip is a fundamental part of every organization and not “just something that circulates within the confines of the organization”. Baumeister et al. [68] have also discussed gossip as a mechanism for organizational learning with multiple functions, including information gathering, disseminating and validating, social enjoyment, and protecting one’s group against norm violators [7,13,16,69]. Gossiping with colleagues can be beneficial in at least three ways on an individual level. Firstly, it is more economical in terms of time and resources to share information with a person who already understands the wider context—as opposed to gossiping with non-work friends. Secondly, gossiping generates reciprocal feelings of belonging and closeness when sharing information with colleagues [54], via its function as a bonding mechanism for social groups [13]. Thirdly, sharing sensitive information helps engender meaningful relationships beyond those dictated by the work role and fulfils the need for a “work confidant/work buddy” [70].

These three benefits suggest that gossip can be a pathway to meeting certain interpersonal and social needs in the workplace that formal communication pathways cannot adequately satisfy. When we view gossip through the lens of demanding healthcare environments, the importance of these benefits can be highlighted. The psychological and social needs of healthcare employees are most often neglected under the pressures of the day-to-day delivery of care, and employee well-being is not an operational priority for healthcare organizations and healthcare management. For example, during the COVID-19 pandemic, the Royal College of Nursing (RCN) and the British Medical Association (BMA) indicated unprecedented numbers of employees leaving, or considering leaving the field of healthcare, with psychological distress and burnout cited as main contributors [71,72,73].

Information in organizations is critical, and access to it is valuable. When information is not transparently shared using the formal channels of communication or not accessible to those who need it [10,74], employees will turn to informal routes, i.e., gossip [7]. Thus, gossip—in forms that do not contain false rumors, bullying intentions, or creating a toxic work environment—is a legitimate form of information exchange which requires for trust to be established to gain access to otherwise confidential information [70]. This means that the receivers of gossip information are considered ingroup members. We can see that the gossip content and the recipients of the information are chosen carefully, in the sense that the appropriate information is shared with the relevant people either because they will share the same opinions with the sender or they will be interested in the sender’s evaluative comments [19]. In this sense, gossip may also contribute to workplace friendship and solidarity, especially when sharing sensitive information [14,70].

In organizations with a high power distance [75] and strict hierarchies [76], such as in healthcare, gossip can also protect employees from feeling powerless at a team level. Gossiping about toxic leaders, for example, has been identified to reduce the possibility of suffering from victimization [77]. When employees gossip about their leaders, they often feel better as gossip helps mitigate the effects of a bad leader’s behavior; when the relationships between the employees are frequent and trusting, more negative gossip about the boss occurs [70]. Gossiping about the “boss” has been linked to cynical beliefs directed at supervisors, as well as the ways they make their decisions [78,79].

In the same vein, gossiping about a toxic leader can warn others or undermine the authority of poor managers [5], as it can function as a way of gathering reputational information that only direct observation allows [80,81]. Feldman [82] argued that gossip rests at the nexus of organizational power and politics. Kurland and Pelled [15] maintained that gossip is a means of acquiring power, highlighting the gain potential for groups lower in the hierarchy. Gossip in organizations is threatening to managers because it is almost impossible to control, as it flows via informal communication channels. Thus, such data are likely to be missed or ignored in AEs, but regularly feature as important in the official inquiries that follow tragic AEs. Healthcare organizations neglect gossip as a potential source of information regarding what is going on in the workplace, as well as the possibility that gossip is connected to the disclosure of important information. For example, in the Kerr/Haslam Inquiry, the investigation concerning the sexual abuse of psychiatric patients revealed that knowledge of these existed in informal channels of communication for over 20 years [4].

### 5.2. Challenge Point 2: Informal Communication Is Not Just “Cheap Talk”—It Is Part of the Group-Level Sensemaking Process

Informal communication, in the form of gossip, is pervasive in healthcare. As a phenomenon that occurs daily, gossip is part of daily work life, often mentioned as a “problem” in healthcare—especially due to the consequences of rumor and malicious gossip [4]. However, evidence suggests that not all gossip is linked to negative outcomes and that valence matters [14,45,83]. The full extent of gossip’s benefits and whether they are more valuable to the workplace than its “dark side” has not been sufficiently explored and, despite the assumption that gossip is pervasive in healthcare, it has not been studied sufficiently concerning other prevailing organizational manifestations, such as burnout, work stress, or quality of care outcomes. We could speculate that formal communication in healthcare—which refers to scheduled events where the staff meets to discuss important topics, usually with a planned agenda (e.g., ward rounds, debriefings, or scheduled appointments with patients and their families) [2]—might be easier to research as opposed to informal communication.

Healthcare contexts have qualities and characteristics that are relatively stable and common across settings and countries [84]. Healthcare organizations thrive on gossip [4] due to complicated interpersonal dynamics within and across teams of diverse people [85]. Congruently, the need to gossip exists because many aspects are not discussed formally. The ubiquity of silence in healthcare sectors [86,87] suggests that there is a significant amount of information that never reaches the formal channels of communication. In times of crisis, informal conversations among trusted staff members are often related to “what is going wrong”; these conversations are a form of group-level sensemaking. Because organizational sensemaking is primarily a social activity, employees will check their interpretations via interacting with others, and then set the norms to act collectively [88]. Gossip is integral to sensemaking, which can lay the groundwork for sensegiving. In their role as sensegivers, line managers in health care have a symbolic role that goes beyond merely expressing values, and symbolic constructions are instrumental in creating meaning for others [89,90]. In sensemaking, gossip can inform new entrants to a profession about standards and appropriate behaviors expected at work, outline injunctive norms (i.e., rules for how not to behave), and serve as a model for well-established norms and regulations.

Gossip is strongly influenced by the context [91]. It emerges over the history of an organization [13], where it can help to establish and maintain important group norms and values [13,68]. In healthcare, it is critical that the context is taken into account, as, for example, regular handover meetings between shifts where formal and informal information is shared between colleagues [92]. Studies of conversations between colleagues are rare, but the existing data suggest that almost 50% of gossip is neither positive nor negative, but neutral [54,93]. Using neutral gossip adds weight to the idea that gossip serves important social functions beyond just information sharing. The COVID-19 pandemic resulted in the use of hybrid meetings, and we have little data yet on whether this hindered gossip. There is some initial evidence that informal communication is more likely to be neglected in remote work settings [94]. The lack of small talk afforded by virtual communication may have negatively impacted well-being, but this is an avenue for future research.

### 5.3. Challenge Point 3: The Role of Informal Communication in “Breaking the Silence” around Critical and Adverse Events

Informal communication will ramp up as crises develop, as individuals seek to make sense of a situation and line managers are under greater pressure to be sensegivers [95]. Within healthcare, gossip and rumor are attempts to process what is happening; as noted by Weick [20], “[o]rganizations talk in order to discover what they are saying, act in order to discover what they are doing”. (p. 191). Congruently, although the content of gossip seems like an important factor, it is the context in which it occurs that is, in fact, the key [96]. Using informal chats fits the literature on voice cultivation within health care [97]. Underestimating gossip runs the risk of missing out on important sources of “soft intelligence” in health care. For example, Weick and Sutcliffe [20] highlight how, in the case of the Bristol Royal Infirmary, there was a continuation of a pediatric cardiac surgery program for almost 14 years despite the data showing a mortality rate roughly double the rate of any other center in England. We can only speculate as to the degree to which gossip among ancillary staff members had identified a serious problem, contrasted with the reported secrecy about doctors’ performance and a lack of monitoring by management. The role of gossip in the extending or shortening of AEs is underresearched, but it is reasonable to suggest that instances of gossip may represent the early rumblings prior to the storms that accompany whistleblowing [1].

As noted by Leary [37], a workforce that must resort to whistleblowing is a symptom of a poor safety culture. Depending on whistleblowing to solve AEs is akin to waiting for a poorly functioning machine to explode. The formal silence that is characteristic of AEs finds expressions in informal routes. The interest in employee silence is entangled in the whistleblowing literature, as the latter represents the idea that a crisis point has been reached for the leadership and management in healthcare. Research and policies have focused on whistleblowing over the past four decades (for a literature review, see Blenkisopp et al. [98]). However, the interest in employee silence as a response to wrongdoings and patient-safety failures is much more recent. We know that employee silence in healthcare will likely go undetected until a breaking point is reached, and that it only sometimes results in whistleblowing [99]. Furthermore, most recent evidence highlights that speaking-up interventions in healthcare shows no significant impact on preventing or effectively resolving situations of employee silence [48]. Thus, the evidence suggests that employee silence in healthcare is shaped, maintained, and backed by pervasive “cultural forces” that characterize the healthcare sector across countries and continents. Accordingly, in healthcare organizations, employee silence can be an active form of behavior that can serve as the best strategy in particular situations, given the potential negative consequences of speaking up [100], indicating that treating silence as simply choosing not to speak up or as the opposite of speaking up [101] is too simplistic. Moreover, being silent in certain situations to avoid exposing colleagues could be considered a participative group climate—which is characterized by shared employee perceptions over what behaviors are more protective on the group level [102]. The behaviors mentioned above are likely to occur in healthcare where a culture of non-disclosure is valorized. Those lower in the hierarchy are less assertive and identify a knowledge gap between their superiors and themselves [49]. In essence, the early etiology of AEs in healthcare is characterized by staff talking internally but not sharing and being ignored, until exasperation is reached where the critical gossip finally reaches patients and families until a whistleblower bursts the bubble. Attaining a better balance between formal reporting and encouraging more talking among colleagues has the potential to prevent AEs and improve the quality of care.

In this paper, we have reviewed the role of information sharing in the development of AEs. The challenge for future research is (1) delineating how the soft intelligence provided via informal communication can be collected and used to prevent AEs, (2) identifying ways to better balance the inverse relationship between formalizing communication and defensive silence among staff, and (3) examining whether informal communication can be used to generate solutions to deal with the ubiquitous problem of employee silence.

## 6. Conclusions

Informal communication is important in healthcare. It serves important functions at the individual, group, and organizational levels. The ongoing movement within healthcare towards standardizing procedures and digitalizing communications within healthcare is likely to increase, and, thus, further overburden healthcare workers, who will seek alternative ways to feel in control of their narratives and safely share sensitive information about work practices. Informal communication can take many forms, but the phenomenon of gossip is worthy of special attention, given that it can allow staff to detect errors, reveal safety issues, and expose unethical behavior [103]. From a motivational perspective, gossiping can fulfil the self-determination theory’s main elements [104], by making individuals feel more competent, in control, and more socially related to their colleagues. Doing so is consistent with a recent integrative review of workplace gossip, demonstrating that gossip serves four specific functions: information exchange, ego enhancement, social integration, and social segregation [105].

Gossip is worthy of greater consideration as an early-warning indicator of serious dysfunction in a healthcare organization. The reluctance of staff to share information outside their immediate team calls on us to understand why people with access to information feel a sense of togetherness with others who have the same access and a sense of separation from those who do not [106]. For example, improving feelings of psychological safety positively affects team effectiveness in healthcare units. Nevertheless, there is little or no literature on how different types of gossip (e.g., informal chats about events at work versus malicious gossip) influence team psychological safety. Interestingly, recent evidence suggests that the impact of psychological safety is more nuanced than previously assumed, whereby high levels of a psychological safety climate can actually harm the performance of routine tasks [107]. This is a reminder that linear assumptions of more always equaling better (e.g., more formal reporting or more psychological safety) is problematic. As noted by Weick [95], “[t]he veneer of rationality that overlies much talk about organizations tends to minimize the role of random activities”. (p. 194). Line managers have a pivotal role to play in terms of listening for what is being talked about, but are faced with the challenge of how they can be in a position to ensure they hear such information.

Ultimately, we have an interesting tension between work policies that seek to further regulate the formal recording of work practices and procedures in contrast with the need for individuals to increase feelings of autonomy via informal communication paths such as gossip. We grow up with the admonishment that ‘you shouldn’t gossip’, but maybe it is time for our healthcare leaders to change that to ‘What are you gossiping about?’.

## Data Availability

All data used are found within the manuscript.

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
