# Peer review of "Why Talking Is Not Cheap: Adverse Events and Informal Communication"

_healthcare, 2024, doi:10.3390/healthcare12060635_

Round 1

Reviewer 1 Report

Comments and Suggestions for Authors

Dear colleagues!

In general, you touched on an interesting topic, which, unfortunately, has a low level of evidence, but at the same time, negatively controls the atmosphere in the team.

1. Unfortunately, it is difficult to objectively and correctly assess the degree of Informal communication: here, allegorically, “advocatus diaboli” is needed as a regulator of reliability

2. First of all, you need a section on evaluation criteria, ranking, inclusion and non-inclusion. In addition, since this is a review, time limits, bibliographic databases, ethical committee approval, and sampling must be specified. It is also appropriate to describe the design of your methodological search.

3. I suggest you make a graphic abstract or diagram using AE - this will facilitate the reader’s understanding and expand the application capabilities of your review.

4. The Covid-19 pandemic has clearly demonstrated a paradigm shift in informal communication: I think it’s right to focus on this.

In general, it is difficult for me to evaluate the list of references, because... You do not indicate the time frame for searching for sources. But I note that you need to add more relevant links, taking into account the points that I wrote above.

Author Response

We would like to thank the reviewer 1 for their comments. We have addressed each comment and provided our response.

  1. In general, you touched on an interesting topic, which, unfortunately, has a low level of evidence, but at the same time, negatively controls the atmosphere in the team.

Our response: We would like to thank the reviewer for recognising the potential contribution of our paper.

  1. Unfortunately, it is difficult to objectively and correctly assess the degree of Informal communication: here, allegorically, “advocatus diaboli” is needed as a regulator of reliability. First of all, you need a section on evaluation criteria, ranking, inclusion and non-inclusion. In addition, since this is a review, time limits, bibliographic databases, ethical committee approval, and sampling must be specified. It is also appropriate to describe the design of your methodological search.

Our response: Our paper was not a systematic review. The category of ‘review’ was the closet category available (in the journal portal) to our type of paper that we submitted to the special issue. We apologise that the this may have caused some confusion for the reviewer. Our paper is a critical review paper and was an invited paper from the editors of the special issue. The format that we adopted was similar to two previous critical review papers authored by the first author of this paper.

Montgomery, A., et al (2023). Employee silence in health care: Charting new avenues for leadership and management. Health Care Management Review, 48(1), 52-60.

Montgomery, A., Panagopoulou, E., Esmail, A., Richards, T., & Maslach, C. (2019). Burnout in healthcare: the case for organisational change. British Medical Journal  2019; 366 doi: https://doi.org/10.1136/bmj.l4774

Thus, the objective of our paper was to identify the gap that exists between two parallel lines of research, namely informal communication and Adverse events. In this sense, the topic of our critical review was not amenable to a systematic review approach. Such papers are typically classified as ‘Analysis’ or ‘Viewpoint’ papers in journals.  The aim of our paper was to present a perspective or viewpoint on a controversial issue related to patient safety. Our three critical challenge points were aimed at adding new arguments to the debate and presented a new perspective, by appropriately drawing on the existing international literature. We hope that we have explained clearly why our paper does not need to conform to the methodologies used in systematic reviews.

  1. I suggest you make a graphic abstract or diagram using AE - this will facilitate the reader’s understanding and expand the application capabilities of your review.

Our response: We are not clear what type of diagram you are proposing. We would welcome more information on what you are suggesting.

  1. The Covid-19 pandemic has clearly demonstrated a paradigm shift in informal communication: I think it’s right to focus on this.

Our comment: Thank you for this comment.

  1. In general, it is difficult for me to evaluate the list of references, because... You do not indicate the time frame for searching for sources. But I note that you need to add more relevant links, taking into account the points that I wrote above.

Our response: We hope we have clarified to your satisfaction why it’s not appropriate for us to address our paper as a systematic review that would need a detailed methodology. Our paper is a viewpoint paper.

We really enjoyed reading all your interesting points, and we hope we have managed to address them to your satisfaction.

Reviewer 2 Report

Comments and Suggestions for Authors

I read with interest the paper titled "Why talking is not cheap: Adverse Events and Informal Communication". The main aspect of the "review" is that is more an opinion paper (viewpoint paper as stated by the authors), than a narrative / scoping /systematic review on the topic. However, the paper is well written and I have some minor comments to add:

1 - Abstract - Abstract is sometimes written in future. Please rephrase "will be"...

2 - I haven't found the explanation of the source of those adverse events. Are them on therapy, on care, on techniques??? Or you present adverse events in a broader way? Please describe it on the paper. 

Please develop on how effective is informal communication in preventing adverse events in healthcare settings compared to formal communication. What the lack of effectiveness on formal communication by healthcare professionals? Why informal communication is some important between patients and healthcare professionals in a setting where formal communication (of diseases, treatment, etc) is fundamental?

 What is the impacts that informal communication can have on patient safety? Are these impacts quantified or measured in a meaningful way?

Can informal communication be described or documented in some way, as protocols? I understand the good benefit of that, but in the end informal communication depends more on the person that is communicating, so how can we say that 100 practitioners/nurses could be equally effective in informal communication. Please discuss.

Author Response

We would like to thank the reviewer 2 for their comments. We have addressed each comment and provided our response.

  1. I read with interest the paper titled "Why talking is not cheap: Adverse Events and Informal Communication". The main aspect of the "review" is that is more an opinion paper (viewpoint paper as stated by the authors), than a narrative / scoping /systematic review on the topic. However, the paper is well written and I have some minor comments to add:

Our response: The category of ‘review’ was the closet category available (in the journal portal) to our type of paper that we submitted to the special issue. We apologise that  this may have caused some confusion for the reviewer. We agree with the reviewer that our paper is a Viewpoint/critical review paper. It was an invited paper from the editors of the special issue.

  1. Abstract - Abstract is sometimes written in future. Please rephrase "will be"...

Our response: Thank you. We have corrected the Abstract.

  1. I haven't found the explanation of the source of those adverse events. Are them on therapy, on care, on techniques??? Or you present adverse events in a broader way? Please describe it on the paper.

Our response: We have provided a definition of Adverse Events at the beginning of section 2. Here is our definition; “An Adverse Event (AE) is an unintended or unexpected incident that causes harm to a patient and may lead to temporary or permanent disability.”

  1. Please develop on how effective is informal communication in preventing adverse events in healthcare settings compared to formal communication. What the lack of effectiveness on formal communication by healthcare professionals? Why informal communication is some important between patients and healthcare professionals in a setting where formal communication (of diseases, treatment, etc) is fundamental?

Our response: There is little research on how effective informal communications can be used in preventing adverse events. In challenge point 1 (section 4.1) we argue that informal communication will become increasingly important as formal ways of reporting increase. We use the example of ‘work arounds’ to highlight how informal ways of working in healthcare are evidence of the importance of the informal. In terms of the ‘why’ we connect our points with the literature on employee silence – which shows that staff are very careful about what they say- as a reaction to the increasing formalisation of communication.

  1. What is the impacts that informal communication can have on patient safety? Are these impacts quantified or measured in a meaningful way?

Our response: This is a really good question. To date, we don’t yet have the research to show the way that informal communication can impact directly on patient safety. One of the arguments in our paper is that analyses of ‘scandals’ indicates that informal communication was the driver towards whistleblowing. Researchers have studied informal communication in many settings, but to date nobody has meaningfully connected it with reduced patient safety.

  1. Can informal communication be described or documented in some way, as protocols? I understand the good benefit of that, but in the end informal communication depends more on the person that is communicating, so how can we say that 100 practitioners/nurses could be equally effective in informal communication. Please discuss.

Our response: We really liked your question. We think you have identified the core problem that we have tried to highlight in our paper. We have gathered enough evidence to show that informal communication channels are important avenues for information sharing, which are set to become even more important as formal communication channels increase. However, as you note in your question, this is not something can be easily captured in a protocol. We think we have gone some way to answering your question via our Challenge points 2 and 3. Challenge point 3 suggests that clinical leaders need to recognise their role as sensemakers and sensegivers, which will go some way to capturing this important soft intelligence in healthcare organisations. The paradox is that providing more ‘oxygen’ for informal communication channels maybe beneficial in helping to prevent adverse events from the bottom-up.

We really enjoyed reading all your interesting points, and we hope we have managed to address them to your satisfaction.

Reviewer 3 Report

Comments and Suggestions for Authors

Dear authors,

I think that area of your research that you presented in this manuscript would be interesting for international audience and beneficial for healthcare system However there is some important issues you have to address.  Your topic don not reflect your work especially part ''Why talking is not cheap'' I suggest to revise. In abstracts please do not use abbreviations, and also do not use references. In introduction in line 35 please add reference for this statement.

You may use Prisma chart for literature review. 

I don't see clear goal what you try to achieve with your work, do you want informal communication to become more formal, if yes whats the instruments. 

Conclusion is too extensive and has parts do not belong there such as line 450-454. You have to be more concise in abstract.

Author Response

We would like to thank the reviewer 3 for their comments. We have addressed each comment and provided our response.

  1. I think that area of your research that you presented in this manuscript would be interesting for international audience and beneficial for healthcare system

Our response: We thank the reviewer for this positive feedback.

  1. However there is some important issues you have to address. Your topic don not reflect your work especially part ''Why talking is not cheap'' I suggest to revise.

Our response: We apologise if we have not understood the point of the reviewer with regard to our title. The choice of the phrase ‘Why talking is not cheap’ was chosen as this is a very common expression in English (i.e., talk is cheap) that reflects the idea that we underestimate the importance of talking, especially informal chats. We believe that it fits very well with our themes around informal communication and gossip. That said, please inform us if we have not 100% understood your point.

  1. In abstracts please do not use abbreviations, and also do not use references. In introduction in line 35 please add reference for this statement.

Our response: We have included one Abbreviation in the abstract (AE = adverse event). We have explained the abbreviation at the beginning of the abstract. We think this helps the reader, but we are happy to spell out in full every time it’s used, if needed. We have not included any references in the abstract.

  1. In introduction in line 35 please add reference for this statement.

Our response: We think this is self-evident and follows logically from the previous line where we name the list of adverse events that we will review. Every review of any healthcare scandal has recommended more formal reporting as a solution to preventing future problems. We hope you will agree that our readers will accept this as an acceptable statement.

  1. You may use Prisma chart for literature review.

Our response: Our paper was not a systematic review. The category of ‘review’ was the closet category available (in the journal portal) to our type of paper that we submitted to the special issue. We apologise that the this may have caused some confusion for the reviewer. Our paper is a critical review paper and was an invited paper from the editors of the special issue. The format that we adopted was similar to two previous critical review papers authored by the first author of this paper.

Montgomery, A., et al (2023). Employee silence in health care: Charting new avenues for leadership and management. Health Care Management Review, 48(1), 52-60.

Montgomery, A., Panagopoulou, E., Esmail, A., Richards, T., & Maslach, C. (2019). Burnout in healthcare: the case for organisational change. British Medical Journal  2019; 366 doi: https://doi.org/10.1136/bmj.l4774

Thus, the objective of our paper was to identify the gap that exists between two parallel lines of research, namely informal communication and Adverse events. In this sense, the topic of our critical review was not amenable to a systematic review approach. Such papers are typically classified as ‘Analysis’ or ‘Viewpoint’ papers in journals.  The aim of our paper was to present a perspective or viewpoint on a controversial issue related to patient safety. Our three critical challenge points were aimed at adding new arguments to the debate and presented a new perspective, by appropriately drawing on the existing international literature. We hope that we have explained clearly why our paper does not need to conform to the methodologies used in systematic reviews.

  1. I don't see clear goal what you try to achieve with your work, do you want informal communication to become more formal, if yes whats the instruments.

Our response: This is a good point. It’s not clear how we can better capture informal communication in healthcare settings. We think you have identified the core problem that we have tried to highlight in our paper. We have gathered enough evidence to show that informal communication channels are important avenues for information sharing, which are set to become even more important as formal communication channels increase. However, as you note in your question, this is not something that can be easily captured in a protocol. We think we go some way to answering your question in our Challenge points 2 and 3. Challenge point 3 suggests that clinical leaders need to recognise their role as sensemakers and sensegivers, which will go some way to capturing this important soft intelligence in healthcare organisations. The paradox is that providing more ‘oxygen’ for informal communication channels maybe beneficial in helping to prevent adverse events from the bottom-up

  1. Conclusion is too extensive and has parts do not belong there such as line 450-454.

Our response: Lines 450-454 refer to our recommendations for future research. In this regard, we thought that such recommendations would be very desirable for the reader and the paper. We apologise if we have not 100% understood your point. Could you please provide more detail in why you want us to delete lines 450-454?

We really enjoyed reading all your interesting points, and we hope we have managed to address them to your satisfaction.

Round 2

Reviewer 1 Report

Comments and Suggestions for Authors

Hello, dear colleagues!

If you write that your article is "Our paper is a critical review paper and was an invited paper from the editors of the special issue. The format that we adopted was similar to two previous critical review papers authored by the first author of this paper ", then why is there no time scale for selecting sources in materials and methods?

Overall, the study design is poorly presented and needs more detail.

I keep my question about the relevance of bibliographic references, since the period of the literature search is not defined. I would like to draw special attention to the lack of data associated with both pre- and post-Covid types of communication.

Author Response

Dear Reviewer

There seems to be some confusion as to the type of paper that we have selected.

We have submitted a viewpoint paper, not a systematic review. 

We would welcome any comments on the arguments that we present in our paper. 

Reviewer 2 Report

Comments and Suggestions for Authors

The authors considered the comments made by the reviewer. I have no further comments to add. My decision is to accept. 

Author Response

Thank you for your acceptance of the paper. 

Reviewer 3 Report

Comments and Suggestions for Authors

Dear authors,

when I suggested to revise title I did it in best intention to show that you are writing for scientific journal and that title has to brief and clear without phrases. 

Abbreviations in scientific articles should start form introduction section. It is not role of author to explain to reviewer why is something useful for the readers.

Its not usual that authors do not address comments and asking additional questions. 

As example, future research belongs to the end of discussion section and not to conclusion.

I would like to ask you again to address comments properly.

Author Response

Dear Reviewer

We will address all the issues  raised. Please see our response to each of your concerns. We have uploaded an updated copy of the manuscript.

  1. when I suggested to revise title I did it in best intention to show that you are writing for scientific journal and that title has to brief and clear without phrases.

Our response: Many scientific papers use phrases in their title. We don’t think it’s inappropriate to use a phrase that helps the reader appreciate the essence of the paper. That said, we would be very happy to hear any suggestions that you have for an alternative title.

  1. Abbreviations in scientific articles should start form introduction section. It is not role of author to explain to reviewer why is something useful for the readers.

Our response: We have removed abbreviations form the abstract and included them at the start of the introduction.

  1. Its not usual that authors do not address comments and asking additional questions.

Our response: We have addressed all comments, and we think its appropriate to ask questions to clarify the comments of the reviewer.

  1. As example, future research belongs to the end of discussion section and not to conclusion.

Our response: We have moved these lines to the end of the discussion as recommended.

Yours

Anthony Montgomery

Olga Lainidi

Katerina Georganta